# Evolution in Real-World Therapeutic Strategies for HIV Treatment: A Retrospective Study in Southern Italy, 2014–2020

**DOI:** 10.3390/jcm11010161

**Published:** 2021-12-29

**Authors:** Nunzia Papa, Simona Cammarota, Anna Citarella, Luigi Atripaldi, Francesca F. Bernardi, Marianna Fogliasecca, Nello Giugliano, Ugo Trama, Micaela Spatarella

**Affiliations:** 1Cotugno Hospital, AORN Ospedali dei Colli, 80131 Naples, Italy; nunzia.papa@ospedalideicolli.it (N.P.); luigi.atripaldi@ospedalideicolli.it (L.A.); micaela.spatarella@ospedalideicolli.it (M.S.); 2LinkHealth Health Economics, Outcomes & Epidemiology S.R.L., 80143 Naples, Italy; anna.citarella@linkhealth.it (A.C.); marianna.fogliasecca@linkhealth.it (M.F.); 3Pharmacy Unit, “Luigi Vanvitelli” University Hospital, 80138 Naples, Italy; bernardi.francesca.futura@gmail.com; 4General Direction of Health Care & Regional Health System Coordination, Drug & Device Politics, Campania Region, 80143 Naples, Italy; ugo.trama@regione.campania.it; 5Department of Experimental Medicine, University of Campania “Luigi Vanvitelli”, 80138 Naples, Italy; giuglianonello92@hotmail.it

**Keywords:** HIV, guidelines, antiretroviral therapy, ART, treatment modification, switching pattern

## Abstract

Changes in HIV treatment guidelines over the last two decades reflect the evolving challenges in this field. Our study examined treatment change patterns throughout a 7-year period in a large Italian cohort of HIV patients as well as the reasons and direction of changes. Treatment-naïve and -experienced HIV patients managed by Cotugno Hospital of Naples between 2014 and 2020 were analyzed. During the period, the proportion of single-tablet regimen treatment sharply increased for the naïve and experienced patients. Regimens containing integrase strand transfer inhibitors rapidly replaced those containing protease inhibitor and non-nucleoside reverse transcriptase inhibitors. The use of the tenofovir alafenamide fumarate/emtricitabine backbone increased rapidly after its introduction in the Italian pharmaceutical market, making up 63.7 and 54.9% of all treatments in naïve and experienced patients, respectively, in 2020. The main reason for treatment changes was optimization and/or simplification (90.6% in 2018; 85.3% in 2019; 95.5 in 2020) followed by adverse effects and virological failure. Our real-world analysis revealed that the majority of treatment-naïve and treatment-experienced patients received antiretroviral drugs listed as preferred/recommended in current recommendations. Regimen optimization and/or simplification is a leading cause of treatment modification, while virologic failure or adverse effects are less likely reasons for modification in the current treatment landscape.

## 1. Introduction

The human immunodeficiency virus (HIV) pandemic continues to be a major global public health issue. In 2020, there were approximately 37.7 million people living with HIV (PLWH), and around 1.5 million people acquired HIV worldwide [1]. Antiretroviral therapy (ART) has transformed the outlook for PLWH, yielding near-normal life expectancy and better quality of life [2]. ART is now recommended for all PLWH, irrespective of their immune status, in order to prevent HIV-related morbidity, mortality, and reduce transmission to others [3,4,5]. Changes in HIV treatment guidelines over the last two decades reflect the evolving challenges in this field. This underscores the importance of considering the content and revisions of treatment recommendations when analyzing time trends of ART, but such studies are rare in the literature.

Over the past few years, several new drugs with improved efficacy, better tolerability and toxicity profiles, and more convenient formulations have become available [6]. For treatment-naïve PLWH, current guidelines continue to recommend the use of a three-drug combination antiretroviral regimen (ART) that combines two nucleoside reverse transcriptase inhibitors (NRTIs) with a third agent (non-nucleoside reverse transcriptase inhibitors [NNRTIs], protease inhibitors [PIs], or integrase strand transfer inhibitors [INSTIs]) [3,4,5,7]. However, the European AIDS Clinical Society (EACS) guidelines are the first to include a two-drug regimen, dolutegravir (DTG) plus lamivudine (3TC), as a recommended first-line treatment option [8]. Moreover, treatment regimens of up to 20 pills per day have been largely replaced by once-daily, single-tablet regimens (STRs) over time [9]. Current international guidelines also recommend that providers, when choosing between regimens of similar efficacy and tolerability, use once-daily (OD) regimens for treatment-naive patients beginning ART, switch treatment-experienced patients receiving complex or poorly tolerated regimens to OD regimens, and use fixed-dose combinations (FDCs) and STRs to reduce pill burden. However, there are limited data on reasons to modify ART regimens in routine clinical care.

In Italy, approximately 90% of PLWH starting HIV treatment achieve and maintain viral suppression [10,11] People living with HIV are exclusively managed by infectious disease specialists, and the ‘test and treat’ approach is often followed, with patients initiating treatment within 3–5 days in many cases [12]. Recent observations suggest that access to care for PLWH can be seriously hindered by shifts in the clinical efforts of HIV specialists towards patients with COVID-19, which spread dramatically in Italy after March 2020 [13]. Drug utilization studies are powerful exploratory tools to evaluate the evolution of therapeutic strategies over time, to assess compliance with national guidelines, and help to better understand the consequences of a critical period on access to care.

The purpose of this study is to examine the change in ART regimens for treatment-naive and treatment-experienced patients over the last seven years (2014–2020), which includes the COVID-19 pandemic era, through the analysis of pharmacy data from Cotugno Hospital of Naples (Campania region, Southern Italy). The secondary aim is to explore the reason and direction of treatment changes after the introduction of new drugs and recommendations.

## 2. Methods

This observational, non-interventional, descriptive study was carried out after approval from the institutional ethics committee of Cotugno Hospital. The article does not contain clinical studies, and all patients’ data were fully anonymized and were analyzed retrospectively. For this type of study, formal consent is not required according to current national law from Italian Medicines Agency.

Cotugno Hospital is one of the largest hospitals in Italy. It provides care to ~2500 PLWH per year (2/3 of the overall Campania region). For this retrospective study, data were retrieved from the hospital pharmacy databases containing outpatient medications dispensed by the hospital pharmacy and reimbursed by local health authorities. For each dispensed medication the following information was collected: anonymous patient code, patient sex and date of birth, origin, date of dispensation, Anatomical Therapeutic Chemical (ATC) code, quantity dispensed, dose, and formulation. Data related to adverse drug reactions were collected through standard-of-care operating procedures utilized in a specialty pharmacy setting. This procedure utilized prescription claims software and a clinical assessment management program according to the national network for pharmacovigilance (RNF—Rete Nazionale Farmacovigilanza) [14]. Adverse events were defined according to FDA regulation (http://www.fda.gov/Safety/MedWatch/HowToReport/ucm053087.htm, accessed on 16 November 2021). Furthermore, since 2018 the Cotugno Hospital pharmacy staff have collected the reasons for every ART regimen change (i.e., optimization or drug–drug interactions) before dispensing medications following a structured interview with patients and clinician. For each ART modification, the following information was collected: anonymous patient code, date of dispensation, Anatomical Therapeutic Chemical (ATC) code, quantity dispensed, dose, and formulation. Data related to adverse drug reactions were collected through standard-of-care operating procedures utilized in specialty pharmacy setting. This information was also extracted and analyzed.

The study population comprised all HIV patients 18 years or older who had at least one pharmacy claim for ART between 1 January 2014 and 31 December 2020. We excluded patients who had ≤2 ART claims during that period in order to exclude HIV prophylaxis treatment. For each calendar year, the date of the first ART claim was set as the index date. Further, patients were considered treatment-naïve if they had no ART dispensed by Cotugno Hospital pharmacy during 1 year before the index date, while treatment-experienced had ≥1 claim for ART during that period. 

The Italian health care service provides ART to all the individuals in need and has recommended universal HIV treatment since 2015 regardless of the initial CD4 count [15]. All antiretroviral drugs approved by the European Medicines Agency (EMA) are available and are provided free of charge to PLWH in our country. The timeline of the Italian Medicines Agency (AIFA) approval for the new antiretroviral drugs throughout the study period is depicted in Figure 1. Therefore, the ART regimens were analyzed at index date as follows: 2 NRTIs + a third agent (NRTI, PI or INSTI), DTG + rilpivirine (RPV) or DTG + 3TC, DTG/RPV or DTG/3TC (as FDC recently approved by AIFA), and others. All medications filled within 14 days from the index date were part of the same ART regimen. The NRTI backbones were identified using ATC codes and classified into three categories: (1) abacavir (ABC)/lamivudine (3TC), (2) tenofovir disoproxil fumarate (TDF)/emtricitabine (FTC) and (3) the tenofovir alafenamide fumarate (TAF)/emtricitabine (FTC). The latter was available in Italy since March 2017.

Finally, switching ART regimens was defined as a treatment change in at least one antiretroviral drug in the regimen excluding dose modifications. Reasons for switching were classified as follows: adverse events, virologic failure, drug–drug interactions, treatment optimization and/or simplification.

Chi-square statistics for trend and analysis of variance (ANOVA) were used to compare distributions of categorical variables and continuous variables, respectively. The proportions of ART regimens and nucleoside backbones dispensed on the index date were calculated by year. Switching patterns between drug class or within NRTI backbones were examined by a matrix analysis to show switches ‘from’ (vertical axis) and switches ‘to’ (horizontal axis) for each drug class (i.e., from PI to INSTI) or NRTI backbones (i.e., from TDF/FTC or ABC/3TC to TAF/FTC). Statistical descriptive analysis was performed using SPSS software version 23 SPSS Inc., Chicago, IL, USA with *p* < 0.05 indicating statistical significance.

## 3. Results

### 3.1. Distributions of Patient Characteristics and ART Regimens during 2014–2020

*Treatment-naive**patients.* The baseline characteristics of patients and the most common ART regimens chosen within the study period are shown in Table 1 and Appendix A. They were predominantly male, and almost three-quarters were less than 50 years old at the index date. 

The proportion of STRs used increased from 48.4 to 81.2% over the study period (*p* < 0.0001). TDF/FTC has been the predominant backbone until 2017. After this treatment initiation with TDF/FTC dropped below 10% in 2020, while the use of TAF/FTC increased accounting for 68.4% of all treatment initiations in the last year. The use of ABC/3TC doubled from 10.3 to 27.1% in the early period (2014–2018) but decreased after 2019. With respect to the third antiretroviral agents, the use of PI and NNRTI decreased throughout the period (PI: 39.5–8.5%; NNRTI: 27.2–15.0%), while the use of INSTI rose sharply, accounting for two-thirds of all third agents prescribed in 2020. Finally, DTG-based two-drug regimens accounted below 3% in 2020, for DTG + 3TC and DTG/3TC.

*Treatment-experienced patients*. During the 7-year observation period the number of treatment-experienced patients increased from 1779 (2014) to 2137 (2020). A full description of baseline characteristics and ART used can be found in Table 2 and Appendix A. They were mainly male and the proportion of patients over 50 years old increased significantly over time (from 35.2 to 47.2%; *p* < 0.0001). The proportion of STR increased constantly over time, amounting to 77.3% in 2020. TDF/FTC was the most commonly prescribed NRTI backbone in the early period. After 2018, there was a sharp decline (29.0–2.1%, from 2018 to 2020), while the use of TAF/FTC increased in the same period (30.1–60.1%). With respect to the third agent, PI were the most commonly prescribed in the first two years, followed by NNRTI. Since 2016, PI and NNRTI prescribing decreased, while there was an increase in use of INSTI (from 13.0 to 50.8% during 2016–2020). The use of DTG-based two-drug regimens increased over time, amounting to 2.3% for DTG + RPV or DTG + 3TC in 2020, whereas the proportion of FDC dual therapy was 2.6%.

### 3.2. Switching Analysis

The number of treatment changes totaled 1855 from 2018 to 2020. The main reason reported for modification of treatment regimen was optimization and/or simplification in each year (90.4% in 2018; 84.7% in 2019; 95.1 in 2020) followed by adverse effects and virological failure (Figure 2).

Over 3 years, 29.8% of all switching involved changes between drug class and 40.8% within NRTI backbones.

With regard to changes between drug class, 48.6 and 24.0% of treatment changes were to INSTI-based three-drug regimens and DTG-based two-drug regimens, respectively. Specifically, 80.1 and 55.2% switched from PIs and NNRTIs to INSTIs; 44.1% on INSTIs switched to dual therapy (13.7% to DTG/RPV or DTG/3TC and 30.4% to DTG + RPV or DTG + 3TC); 43.3 and 13.4% from others to INSTIs and dual therapy, respectively (Table 3).

Regards to changes within NRTI backbones, 91.8% of these was to TAF/FTC: 648 switched from TDF/FTC (95.8% of all changes from TDF/FTC) and 72 from ABC/3TC (100% of all changes from ABC/3TC) (Table 4).

## 4. Discussion

This study is a detailed examination of 7 years of temporal trends (2014–2020) in the prescribing pattern of antiretroviral drugs and reasons for ART switches in a large Italian cohort of HIV patients. The dynamic treatment landscape in which an increased number of co-formulated and STR options became available, with evolution of guidelines over the study period, is evident in the changing prescribing trends and reason for switching. Broadly, our real-world analysis of ART patterns reveals that the majority of treatment-naïve and treatment-experienced patients received regimens that were listed as recommended in current guidelines. Regimen optimization and/or simplification is a leading cause of regimen modification; virologic failure or adverse effects are less likely causes in the current treatment landscape. 

First, we found a rapid increase in the use of STRs throughout the study period, amounting to over three-quarters of all ART regimens prescribed in 2020. These findings confirm that one of the most important challenges for clinicians managing HIV infection remains to encourage patients to take antiretroviral drugs correctly for a lifetime. Indeed, the advent of STRs has allowed a significant simplification of ART regimens in most treatment-naïve and -experienced patients. Results of previous surveys show that patients prefer to take fewer daily pills and look for compact easy-to-use regimens; observational and controlled studies indicate that virological and clinical outcomes are better in individuals treated with STRs versus MPRs, even among difficult-to-treat populations [16]. In our study, we found that the main reason for switching of therapy was overwhelmingly due to optimization and/or simplification. These results are in line with other previous studies showing that most changes were not driven by virologic failure or adverse events, but were strategic, including the prevention of future toxicities with new ART options [17]. 

Second, we observed a dramatic shift in prescribing patterns over time, with INSTI-based regimens rapidly replacing PI and NNRTI-based regimens during the later period.

The switching matrix between drug class showed that 80 and 55% of all changes from PIs and NNRTIs (between 2018 and 2020) were towards INSTI-based regimens. The above findings are in accordance with international and national guidelines for ART, which recommend INSTI-based regimens in treatment-naïve and treatment-experienced patients [3,4,5,18]. Several clinical trials and observational studies have demonstrated that INSTIs have a high potency, more favorable safety and tolerability profiles than older agents, and second-generation INSTIs (DTG and BIC) have high genetic barriers to resistance. In contrast, PI-based regimes cause more intolerance/adverse effects, and NNRTI-based regimes have a lower barrier for the development of resistance [19].

The appeal of INSTIs mainly increased after the AIFA’ s approval of additional once-daily STR containing elvitegravir (EVG), dolutegravir (DTG), and the new INSTI bictegravir (BIC), especially in combination with TAF/FTC.

Regarding the NRTI backbones, TDF/FTC was the most commonly used backbone until 2017. After TAF entered in the Italian pharmaceutical market on March 2017, we found that prescribers were more likely to prescribe TAF than TDF, both in treatment-naïve and treatment-experienced PLWH. This may be reflecting the preference among clinicians for TAF, especially in an aging population with comorbidities. In several clinical trials and cohort studies, TAF, given in combination with other antiretroviral agents, has been associated with high efficacy and lower risk of renal and bone toxicities compared with TDF both in treatment-naïve and treatment-experienced patients [20]. In addition, the switching matrix within drug class in this study reported switches from ABC/3TC to TAF/FTC. This switch was usually associated with no difference in renal or bone safety profile and it could be considered in patients with high cardiovascular disease (CVD) risk [21]. A systematic review and meta-analyses of results from 17 epidemiologic studies found a 61% increased pooled risk of CVD among PLWH who were recently exposed to abacavir. In view of this increased risk of CVD, risks and benefits for PLWH must be carefully weighed in prescribing abacavir-based regimens, taking into account existing risk factors for CVD and the patient’s clinical status [22].

In addition, our study also explores prescribing of dual therapy in real-world setting, DTG plus 3TC and DTG plus RPV, recently approved also as FDC by AIFA. Since 2017, Italian guidelines recommend DTG based 2-drug regimens, underlying the efficacy of these regimens in patients harboring variants bearing reverse transcriptase inhibitor resistance mutations or in patients with a history of virological failure to reverse transcriptase inhibitors [23]. In 2019, the EACS guidelines were the first to introduce a two-drug regimen (DTG plus 3TC) as a recommended first-line treatment option [8]. The increased use of DTG-based 2-drug regimen prescribing may reflect the tendency of clinicians to adopt new treatment paradigms informed by clinical trial results or clinical experience also prior to their addition to the guidelines.

Finally, we observed that the number of treatment-experienced patients remained stable in the pandemic era, suggesting that retention in HIV care has been assured despite COVID-19 restrictions. Hospitals in Italy are to dispense antiviral drugs in 3–6-month doses to meet the needs of people living with HIV and reduce facility visits [12]. Since the lockdown of Italy on 10 March 2020, a community-based organization has dedicated resources to ensure home delivery of ART medications from our hospital pharmacy through volunteers, Civil Protection, and the Red Cross and to avoid patient exposure to SARS-CoV-2. However, as also shown in other experiences, we observed a drop in the number of naïve-treatment patients in 2020, suggesting that the COVID-19 emergency may have had a negative impact on HIV screening programs [13].

Our study has several limitations. First, this is a retrospective study without follow-up; therefore, it is not possible to ascertain the consequences and the efficacy of prescribing strategies. Second, the viral load and CD4+ count were not reported in this study. Third, no data were available on clinical and psychosocial factors that could have affected ART prescribing. Finally, this analysis was limited to one site, and prescribing patterns in this clinic may not reflect decision-making and management elsewhere in Italy. As most of the physicians at Cotugno Hospital have a specific interest and expertise in HIV care and are involved in the evaluation of new agents through clinical trials or as opinion leaders, they will generally be knowledgeable about new treatments and are more likely to be early adopters of new therapeutic approaches.

In conclusion, the overarching goal of ART is still to achieve and maintain virologic suppression in all patients, but regimen simplification, especially with potent STRs, and optimization of ART to reduce toxicities and ensure long-term tolerability has become an increasingly important goal in HIV management, particularly in the context of an aging population. Taken together, these findings suggest that in the contemporary ART era treatment changes are more likely to occur for optimization rather than as a result of adverse effects and virologic failure.

## Figures and Tables

**Figure 1 jcm-11-00161-f001:**
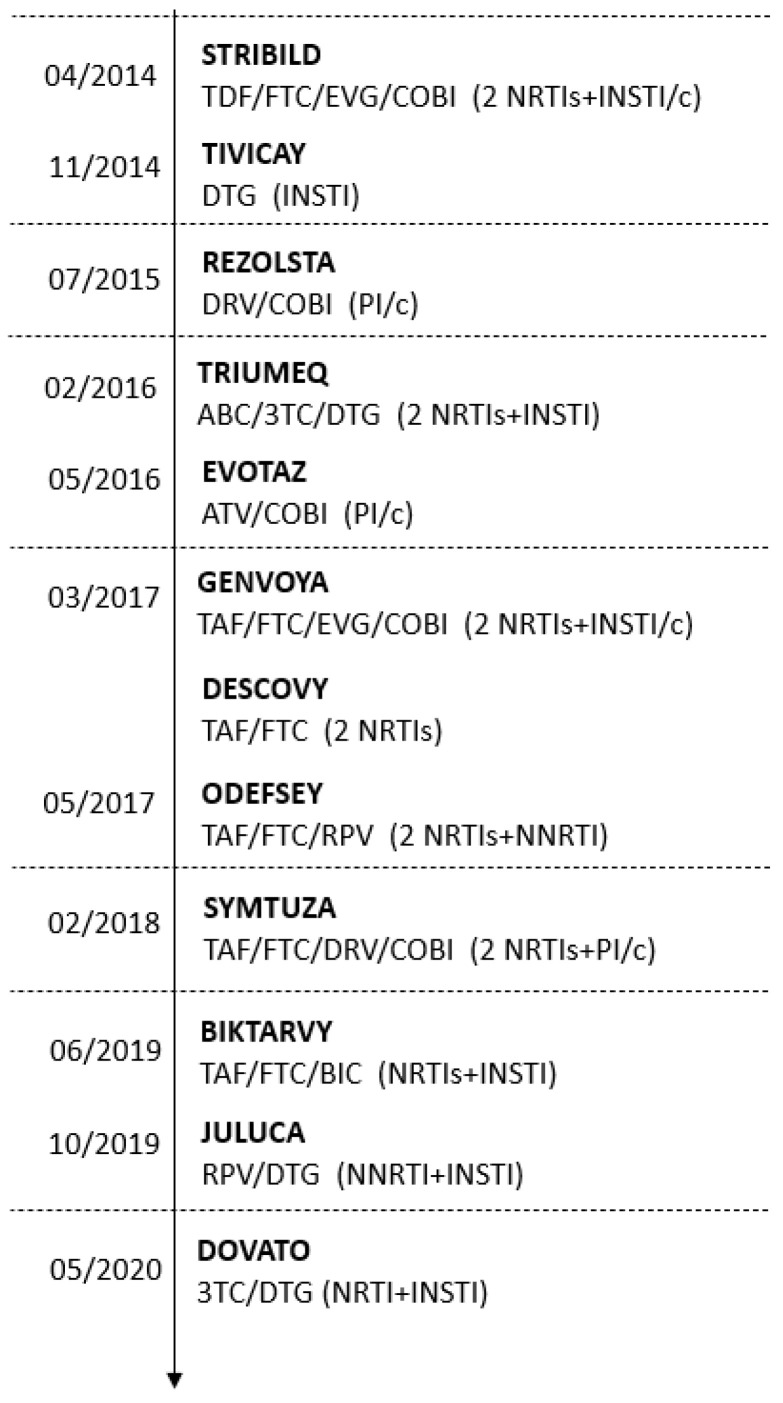
Timeline of the Italian Medicines Agency (AIFA) approval for antiretroviral medications during 2014–2020. Abbreviations: 3TC, lamivudine; ABC, abacavir; ATV, atazanavir; BIC, bictegravir; COBI, cobicistat (used as booster = /c), DRV, darunavir; DTG, dolutegravir; EVG, elvitegravir; FTC, emtricitabine; INSTI, integrase strand transfer inhibitor; NRTI, nucleos(t)ide reverse transcriptase inhibitors; NNRTI, non-nucleoside reverse transcriptase inhibitors; PI/c, protease inhibitor pharmacologically boosted with cobicistat; RPV, rilpivirine; TAF, tenofovir alafenamide; TDF, tenofovir disoproxil fumarate.

**Figure 2 jcm-11-00161-f002:**
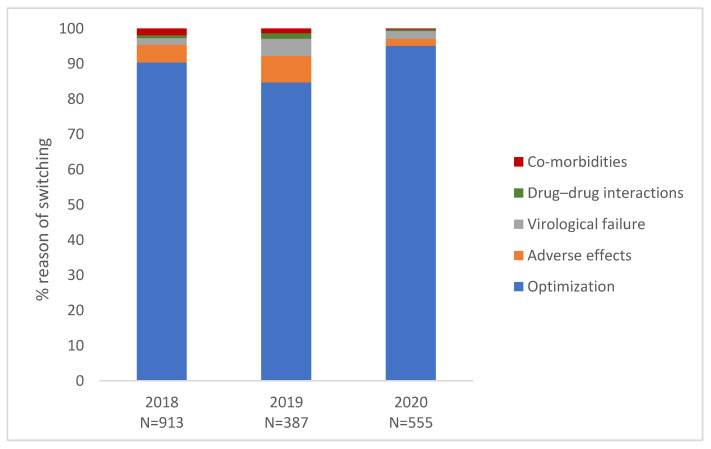
Reason for switching by year (2018–2020).

**Table 1 jcm-11-00161-t001:** Characteristics for treatment-naïve patients (2014–2020).

	2014 *N* = 349	2015 *N* = 375	2016 *N* = 319	2017 *N* = 293	2018 *N* = 292	2019 *N* = 393	2020 *N* = 234	F Value ^1^/ χ^2^ Value (*df*) ^1^	*p* Value
Male (%)	74.8	69.3	75.9	80.9	78.4	71.0	75.6	17.0 (6)	0.009
Age									
Mean (years)	42.7	43.6	42.3	41.8	41.2	41.4	42.4	2.1 (6; 2248)	0.055
≥50 years	26.1	24.0	28.2	25.3	20.2	22.1	24.4	7.1 (6)	0.314
Non-Italian origin (%)	16.3	28.3	21.6	20.8	22.9	17.6	18.8	20.8 (6)	0.002
STR (%)	48.4	44.5	58.6	54.6	58.6	64.1	81.2	99.7 (6)	<0.0001
MTR (%)	51.6	55.5	41.4	45.4	41.4	35.9	18.8		
NRTI backbones (%)									
ABC/3TC	10.3	15.7	22.6	23.5	27.1	18.6	17.1	1361.9 (36)	<0.0001
TDF/FTC	65.0	62.1	61.8	58.0	7.5	3.8	6.0		
TAF/FTC				10.9	57.2	69.0	68.4		
ART regimen (%)									
2 NRTIs + PI	39.5	30.4	19.4	21.8	22.9	20.9	8.5	387.9 (30)	<0.0001
2 NRTIs + NNRTI	27.2	23.2	19.7	18.1	12.3	12.7	15.0		
2 NRTIs + INSTI	10.0	24.8	44.5	49.1	54.5	55.7	67.9		
Others	23.2	21.6	16.3	10.9	9.9	9.2	6.0		
DTG + 3TC ^2^					0.3	1.5	1.3		
DTG/3TC ^2^							1.3		

Abbreviations: 3TC, lamivudine; ABC, abacavir; df, degree of freedom; DTG, dolutegravir; ART, combination antiretroviral treatment; FTC, emtricitabine; INSTI, integrase strand transfer inhibitor; MTR, Multiple Tablet Regimen; NRTI, nucleos(t)ide reverse transcriptase inhibitors; NNRTI, non-nucleoside reverse transcriptase inhibitors; PI, protease inhibitor; STR, Single Tablet Regimen; TAF, tenofovir alafenamide; TDF, tenofovir disoproxil fumarate. ^1^ F value in ANOVA for continuous variables and χ^2^ value in chi-square test for categorical variables. ^2^ The 2-drug regimen recommended by European AIDS Clinical Society (EACS) guidelines (Update 2019) as first-line treatment option.

**Table 2 jcm-11-00161-t002:** Characteristics for treatment-experienced patients during 2014–2020.

	2014 *N* = 1779	2015 *N* = 1935	2016 *N* = 2066	2017 *N* = 2055	2018 *N* = 2093	2019 *N* = 2191	2020 *N* = 2137	F Value/ χ^2^ Value (*df*) ^1^	*p* Value
Male (%)	73.0	73.7	73.9	74.2	75.0	75.0	75.0	3.7 (6)	0.716
Age									
Mean (years)	46.2	46.8	47.2	47.4	47.6	47.8	48.1	7.6 (6; 14249)	<0.0001
≥50 years	35.2	38.5	40.2	42.8	44.7	45.6	47.2	87.6 (6)	<0.0001
Non-Italian origin (%)	16.0	14.6	15.3	15.6	16.2	15.9	14.8	3.5 (6)	0.743
STR (%)	31.5	39.1	43.0	51.5	56.0	60.6	77.3	1123.0 (6)	<0.0001
MTR (%)	68.5	60.9	57.0	48.5	44.0	39.4	22.7		
NRTI backbones (%)									
ABC/3TC	16.5	16.1	17.1	21.4	23.8	24.4	22.7	6868.8 (54)	<0.0001
TDF/FTC	58.5	57.7	58.0	56.6	29.0	4.5	2.1		
TAF/FTC				0.8	30.1	56.3	60.1		
ART regimen (%)									
2 NRTIs + PI	43.7	37.0	29.3	23.8	19.7	17.3	16.5	3100.1 (30)	<0.0001
2 NRTIs + NNRTI	30.6	32.7	31.9	28.1	23.4	20.8	17.1		
2 NRTIs + INSTI	1.5	5.0	13.0	27.6	39.6	46.6	50.8		
Others	24.2	25.2	25.5	19.8	15.7	13.3	10.7		
DTG + RPV or DTG + 3TC		0.1	0.3	0.7	1.6	2.0	2.3		
DTG/RPV or DTG/3TC							2.6		

Abbreviations: 3TC, lamivudine; ABC, abacavir; df, degree of freedom; DTG, dolutegravir; ART, combination antiretroviral treatment; FTC, emtricitabine; INSTI, integrase strand transfer inhibitor; MTR, Multiple Tablet Regimen; NRTI, nucleos(t)ide reverse transcriptase inhibitors; NNRTI, non-nucleoside reverse transcriptase inhibitors; PI, protease inhibitor; RPV, rilpivirine; STR, Single Tablet Regimen; TAF, tenofovir alafenamide; TDF, tenofovir disoproxil fumarate. ^1^ F value in ANOVA for continuous variables and χ^2^ value in chi-square test for categorical variables.

**Table 3 jcm-11-00161-t003:** Switching matrix for ART drug class (2018–2020).

			To					
		Total	2 NRTIs + PI	2 NRTIs + INSTI	2 NRTIs + NNRTI	DTG/RPV or DTG/3TC	DTG + RPV or DTG + 3TC	Others
		*N*	%	%	%	%	%	%
From	2 NRTIs + PI	161	-	80.1	5.6	1.9	3.7	8.7
	2 NRTI + INSTI	102	18.6	-	19.6	13.7	30.4	17.6
	2 NRTIs + NNRTI	145	4.1	55.2	-	4.1	33.8	2.8
	DTG + RPV or DTG + 3TC	8	-	25.0	-	12.5	50.0	12.5
	DTG/RPV or DTG/3TC	4	-	-	50.0	25.0	-	25.0
	Others	134	12.7	43.3	2.2	6.7	6.7	28.4
	Total	554	7.6	48.6	6.1	6.1	17.9	13.7

Abbreviations: 3TC, lamivudine; DTG, dolutegravir; INSTI, integrase strand transfer inhibitor; NRTI, nucleos(t)ide reverse transcriptase inhibitors; NNRTI, non-nucleoside reverse transcriptase inhibitors; PI, protease inhibitor; RPV, rilpivirine.

**Table 4 jcm-11-00161-t004:** Switching matrix for NRTI backbones (2018–2020).

			To			
		Total	TDF/FTC	ABC/3TC	TAF/FTC	TDF/3TC
		*N*	%	%	%	%
From	TDF/FTC	648	-	4.2	95.8	-
	ABC/3TC	72	-	-	100.0	-
	TAF/FTC	33	30.3	66.7	-	3.0
	TDF + 3TC	3	-	-	100.0	-
	ZDV/3TC	2	-	100.0	-	-
	Total	758	1.3	6.7	91.8	0.1

Abbreviations: 3TC, lamivudine; ABC, abacavir; FTC, emtricitabine; NRTI, nucleos(t)ide reverse transcriptase inhibitors; TAF, tenofovir alafenamide; TDF, tenofovir disoproxil fumarate; ZDV, zidovudine.

## Data Availability

Restrictions apply to the availability of these data. The readers may contact the authors to access these data.

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
