# Peer review of "Evolution in Real-World Therapeutic Strategies for HIV Treatment: A Retrospective Study in Southern Italy, 2014–2020"

_jcm, 2021, doi:10.3390/jcm11010161_

Round 1

Reviewer 1 Report

The Manuscript ID: jcm-1499931 entitled “Evolution in the real-world therapeutic strategies for HIV treatment: retrospective study in the Southern Italy, 2014-2020” is a well written manuscript describing the switch in the ART drugs prescriptions in Italy from 2014 to 2020. The authors have shown, based on a single centre hospital, that the main reason of changes in drugs prescriptions are optimization rather than as a result of adverse effects and virologic failure. In this reviewer opinion, the study is scientifically sound, the manuscript is well written and acceptable for publication. I only have a few minor comments:

Specific comments:

1- Please, verify if the word “INSTI” should be written in capital letters in the sentence “…..was carried out after approval from INSTItutional ethics committee of the Cotugno Hospital.”

2- Please, standardize the use of the word “INSTI”. In the legend of table 2 it is not written in capital letters.

Author Response

1. Please, verify if the word “INSTI” should be written in capital letters in the sentence “…..was carried out after approval from INSTItutional ethics committee of the Cotugno Hospital.”

Thanks for the comment. We modified it in the text.

2.Please, standardize the use of the word “INSTI”. In the legend of table 2 it is not written in capital letters.

Thank you. We standardized the use of the word “INSTI” in the manuscript.

Reviewer 2 Report

The manuscript by Papa et al. utilizes a retrospective design to evaluate changes in therapeutic strategies for HIV-1 in Italy. The primary findings of the manuscript were two-fold: 1) Most individuals, including both treatment-naïve and treatment-experienced, receive treatments aligning with the current recommendations; and 2) Treatments are primarily modified for simplification and/or optimization.  Overall, the manuscript is well-written and sheds light on an important topic. The reviewer has a few outstanding questions for consideration.

  1. For treatment-experienced patients, a significant subset of individuals are of non-Italian origin. Is any of the variance in ART regimen attributable to origin?
  2. The percentages for NRTI Backbones in Tables 1 and 2 do not add up to 100%. Are there additional combinations of NRTI Backbones that are commonly used? If so, it may be beneficial to account for these combinations in the tables.
  3. Sex differences in multiple factors associated with HIV-1 (e.g., HAART Treatment: Floridia et al., 2008; Lemly et al., 2009) are well recognized. The reviewer believes an examination of how biological sex influences ART treatment regimen and/or the reasons for switching would be advantageous.
  4. Please provide complete statistical analyses throughout the results section for all results, including F values, and/or χ2 values, degrees of freedom, and p Assessment of measures of effect size would also be helpful.
  5. Figure 2 allows readers to easily and clearly identify the primary reasons for switching ART regimen. It may be beneficial to include Figures (in addition to the Tables) that illustrate the primary findings of the study.

Author Response

1.For treatment-experienced patients, a significant subset of individuals are of non-Italian origin. Is any of the variance in ART regimen attributable to origin?

Thank you for the comment. We observed similar trends both in the Italian and no-Italian origin patients as shown in the following Tables (details see attachment).

The percentages for NRTI Backbones in Tables 1 and 2 do not add up to 100%. Are there additional combinations of NRTI Backbones that are commonly used? If so, it may be beneficial to account for these combinations in the tables.

Thank you for your comment. There were no other NRTI backbones combinations commonly used. All the other combinations are <1% for each year. The ZDV/3TC was >1% but only in the 2014 year (2.3%). The TAF/FTC proportions were updated because incorrectly transcribed in the table included in the manuscript.

3. Sex differences in multiple factors associated with HIV-1 (e.g., HAART Treatment: Floridia et al., 2008; Lemly et al., 2009) are well recognized. The reviewer believes an examination of how biological sex influences ART treatment regimen and/or the reasons for switching would be advantageous.

We agree with you that an examination of the sex differences both for ART treatment regimen and the reasons for switching would be very interesting mainly for the tolerability.

However, we did not include this topic in the manuscript because data related to reasons for ART regimen switching not include the sex information.

4. Please provide complete statistical analyses throughout the results section for all results, including values, and/or χ2values, degrees of freedom, and p Assessment of measures of effect size would also be helpful.

As suggested, we added values and/or χ2 values, degrees of freedom in Table 1 and 2.

5. Figure 2 allows readers to easily and clearly identify the primary reasons for switching ART regimen. It may be beneficial to include Figures (in addition to the Tables) that illustrate the primary findings of the study.

Thanks for your suggestion. We added Supplementary Figure 1 and 2 to illustrate our main findings.
